# Biocompatibility of α-Al_2_O_3_ Ceramic Substrates with Human Neural Precursor Cells

**DOI:** 10.3390/jfb11030065

**Published:** 2020-09-16

**Authors:** Akrivi Asimakopoulou, Ioannis Gkekas, Georgia Kastrinaki, Alessandro Prigione, Vasileios T. Zaspalis, Spyros Petrakis

**Affiliations:** 1Chemical Process & Energy Resources Institute/Centre for Research and Technology Hellas, 57001 Thessaloniki, Greece; 2Institute of Applied Biosciences/Centre for Research and Technology Hellas, 57001 Thessaloniki, Greece; gkekasioannis@certh.gr; 3Laboratory of Inorganic Materials/Chemical Process & Energy Resources Institute/Centre for Research and Technology Hellas, 57001 Thessaloniki, Greece; georgiak@certh.gr (G.K.); zaspalis@cperi.certh.gr (V.T.Z.); 4Department of General Pediatrics, Neonatology and Pediatric Cardiology University Clinic Düsseldorf, Heinrich Heine University, 40225 Düsseldorf, Germany; alessandro.prigione@hhu.de; 5Department of Chemical Engineering, Aristotle University, P.O. Box 1517, 54006 Thessaloniki, Greece

**Keywords:** a-alumina, nanoporosity, wettability, protein adsorption, human neural precursor cells, biocompatibility, differentiation, neuron

## Abstract

Background: Biocompatible materials-topography could be used for the construction of scaffolds allowing the three-dimensional (3D) organization of human stem cells into functional tissue-like structures with a defined architecture. Methods: Structural characterization of an alumina-based substrate was performed through XRD, Brunauer–Emmett–Teller (BET) analysis, scanning electron microscopy (SEM), and wettability measurements. Biocompatibility of the substrate was assessed by measuring the proliferation and differentiation of human neural precursor stem cells (NPCs). Results: α-Al_2_O_3_ is a ceramic material with crystallite size of 40 nm; its surface consists of aggregates in the range of 8–22 μm which forms a rough surface in the microscale with 1–8 μm cavities. The non-calcined material has a surface area of 5.5 m^2^/gr and pore size distribution of 20 nm, which is eliminated in the calcined structure. Thus, the pore network on the surface and the body of the ceramic becomes more water proof, as indicated by wettability measurements. The alumina-based substrate supported the proliferation of human NPCs and their differentiation into functional neurons. Conclusions: Our work indicates the potential use of alumina for the construction of 3D engineered biosystems utilizing human neurons. Such systems may be useful for diagnostic purposes, drug testing, or biotechnological applications.

## 1. Introduction

The nervous system is characterized by high-complexity and three-dimensional (3D) interconnectivity of its cellular components, i.e., the neurons and non-neuronal glial cells. However, in brain disorders (e.g., traumatic brain injury), tissue architecture is damaged and neuronal signals which regulate the homeostasis and function of the body are not properly transmitted. The recent advances in induced-pluripotent stem cell technology and the development of the appropriate differentiation protocols allowed the in vitro generation of human neuronal cells which could be used for biotechnological or regenerative therapeutic approaches. These cells are predominantly grown in two-dimensional (2D) monolayer cultures which are easy to use and analyze. However, tissue-specific architecture and mechanical signals that regulate the maturation, communication, and physiological function of neurons are not properly modelled. In contrast, 3D cultures allow the organization of neuronal cells into tissue-like structures that better reproduce the in vivo microenvironment. Neural blocks [1] or cell spheroids have been previously utilized for the construction of complex 3D structures (reviewed in [2]). Although promising, these structures may not be functional in vitro or in vivo, due to insufficient provision of nutrients in cells of their inner layers or limited integration and poor functionality into the host tissue. On the other hand, scaffolds made from biocompatible materials may provide structural support for cell adhesion, proliferation, and differentiation mimicking the cellular microenvironment. A number of scaffolds made from solid [3,4,5], microfibrous materials [6], or gels [7] have been previously described to support the proliferation and localized differentiation of murine neural stem cells, indicating their potential value for neural tissue regeneration [8]. Among them, 3D foam biomaterials provide efficient cell adhesion, proliferation, and differentiation due to their unique properties (e.g., high surface-to-volume ratio, 3D porous structure) [9]. Such materials may be also used for the construction of engineered biosystems modeling brain compartmentalization that would allow real-time analyses of brain function [10].

Bioceramics (e.g., alumina, zirconia, titania, hydroxyapatite, glass ceramics) are used for implantation, repair, and reconstruction of the diseased or damaged parts of the body [11]. In some cases, they can strongly bond to living tissues, creating a stable interface and triggering a range of biological responses, such as tissue regeneration while degrading over time [12]. In recent years alumina has been preclinically and clinically studied in designing dental and orthopedic biomaterials [13,14,15] while the controlled network of its porous structure provides advanced functionalities to immunoisolation devices, scaffolds for tissue engineering, biomolecular filtration, and state of the art controlled release drugs [16,17,18]. The structural characteristics and composition of alumina substrates make them proper scaffolds for cell cultures, based on the optimized surface wettability for cell adhesion and the surface chemical modification with functional groups which may control the differentiation of neural stem cells [19].

The aim of the present work is to test the biocompatibility of an alumina-based ceramic material which could be fabricated in desired shapes and evaluate whether it supports the proliferation and differentiation of human neural precursor cells (NPCs). A thorough characterization of the surface of non-calcined and calcined alumina was first performed, in order to better understand the cell-material interaction. Most importantly, we show that alumina is biocompatible with human NPCs as it supports their proliferation, differentiation, and survival of functional NPC-derived neurons. These results indicate that alumina may be used for the development of scaffolds for neural-like 3D structures.

## 2. Materials and Methods

### 2.1. Ceramic Substrate Surface Modification

Alumina discs (Τ-ceramics machinable alumina, THERMANSYS^®^) of 11 mm diameter and 4 mm height (suitable to fit in 12-well insert plate) were selected. After full firing to 1700 °C, a dense material with all mechanical, chemical, and electrical properties of high purity recrystallized alumina was obtained (Table 1). Full firing temperature profile is as follows: temperature increase to 1100 °C with a heating rate of 4 °C/min and then to 1700 °C with a heating rate of 2 °C/min, maintaining the temperature at 1700 °C for 6 h and cool down the furnace gradually with a rate of 4 °C/min. A degree of shrinkage of 20–21% approximately was noticed, resulting in reduced dimensions of the final disc (8.8 mm diameter and 3.2 mm height) as well in reduced porosity and pore size. Both non-calcined (ncA) and calcined (cA) samples of Τ-ceramics machinable alumina were structurally characterized via state-of-the-art material surface analysis methods, while only calcined alumina was evaluated as a potential substrate for cell culture.

### 2.2. XRD Spectroscopy

The X-Ray diffraction was performed by an XRD D500/501 apparatus (Siemens, Berlin, Germany), equipped with Cu Ka radiation source from 10–80 2θ angle with 0.04 step.

### 2.3. Brunauer–Emmett–Teller (BET) Analysis

Surface porosity, average pore size and pore size distribution were determined for all alumina samples by BET method employing a Micromeritics Tri-Star system at 433 K and maintained at this temperature for 4 h under vacuum (~50 mTorr) and degassing at 250 °C.

### 2.4. Wettability

The wetting ability of the ceramic material was measured by the Water Contact Angle method. An in-house contact angle measurement instrument based on a high-resolution sensor camera (Nikon D5600, Tokyo, Japan, AF-P NIKKOR 18–55 mm 1:3.5–5.6 G) was employed according to the arrangement proposed in [21]. After cleaning the surface of the ceramics with acetone, a 50 μL double distilled water drop was added manually via a glass Pasteur pipette with opening of 1 mm. Drop images on the surface of each material were taken after 1–2 s. For each material (non-calcined and calcined alumina), three samples were tested and for each sample, three measurements on different locations of their surface were performed. With the use of proper lighting and alignment of the sample/focal point and the camera, photos with no reflection were obtained. The contact point between solid, liquid, and gas phase (i.e., ceramic surface, water drop, and air, respectively) was detected by the intersection between the solid surface line and the drop profile, both detected after digitally zooming the images and processing them properly in a raster graphics editor software.

### 2.5. NPC Culture

NPCs were obtained from healthy human induced pluripotent stem cells (iPSCs) (ethical approval: IRB code #EA2/131/13), as previously described [22]. Calcined Al_2_O_3_ ceramic discs were coated with Matrigel (BD Biosciences). NPCs were seeded on discs (0.25 × 10^6^ cells/cm^2^) and cultured in DMEM-F12/Neurocult 1:1 supplemented with 1:200 N_2_ supplement, 1:100 B27 supplement lacking vitamin A, 3 μM CHIR 99,021 (Sigma-Aldrich, Saint Louis, MO, USA), 0.5 μM purmorphamine, 150 μM ascorbic acid and 1% penicillin/streptomycin/glutamine in a humidified incubator set to 37 °C and 5% CO_2_. For neuronal differentiation, NPCs on ceramic substrates were cultured for 21–35 days in DMEM-F12/Neurocult 1:1 supplemented with 1:200 N_2_ supplement, 1:100 B27 supplement lacking vitamin A 8. All cell-culture media were changed every 48 h.

### 2.6. MTT Cell Viability Assay

For the estimation of cell viability on calcined Al_2_O_3_ ceramic substrates, NPCs were incubated with 5 ug/mL MTT (Millipore) for 4 h at 37 °C and 5% CO_2_. Culture medium was removed and formazan crystals were solubilized with DMSO. OD 570/630 was measured in a BioTek ELX800 apparatus (BioTek, Winooski, VT, USA).

### 2.7. RT-qPCR

Total RNA was purified from control or differentiated NPCs grown on ceramic substrates using the NucleoSpin RNA kit (Macherey-Nagel, Düren, Germany), according to manufacturer’s instructions. cDNA generation and RT-qPCR reactions were performed using the KAPA SYBR FAST One-step Universal kit (KAPA Biosystems) in a Rotor-Gene 6000 operating system (Qiagen, Hilden, Germany). Primer sequences are shown in Table 2. The correct size of amplified RT-qPCR products was verified by electrophoresis in a 2% agarose gel.

### 2.8. Statistical Analysis

Statistical analysis was performed using the GraphPad Prism software v.4 (San Diego, CA, USA). All experiments were performed in triplicates and results are shown as mean ± SD calculated by a T-test.

### 2.9. Scanning Electron Microscopy (SEM)

NPCs on ceramic substrates were fixed with 4% glutaraldehyde for 30 min before dehydrated in increasing concentrations of ethanol. Samples were air dried, mounted on SEM stubs, using a two-sided adhesive film, and covered by a thin gold coating. Microscopic observation was carried out using a JSM-6300 JEOL SEM instrument (JEOL Ltd., Tokyo, Japan), operating at an accelerating voltage of 20 kV.

## 3. Results

### 3.1. Substrate Characterization

The properties of the substrate material, such as surface microstructure and chemical composition as well as mechanical properties, have a significant effect on the bioactivity of cells and the construction of building blocks for engineered biosystems [19,23]. Cell adhesion, survival, and proliferation on materials largely depends on surface characteristics such as wettability, chemistry, charge, rigidity, and roughness.

#### 3.1.1. Surface Topology and Nanoporosity

Alumina samples (Table 3) were characterized via X-Ray diffraction spectroscopy for their crystal structure and by BET analysis for their surface area calculation and pore size distribution profile, where measurable. In the case of cA, the surface area corresponded to a dense structure and could not be measured by the BET analysis. Figure 1 depicts the hexagonal crystal structure of α-Al_2_O_3_ and a proposed mechanism for protein attachment on the surface hydroxyl groups that are formed when alumina exists in aquatic environments. FTIR spectroscopy performed on the cA and ncA samples (Appendix A) validated the presence of characteristic surface hydroxyl groups on which proteins may attach.

##### XRD Spectroscopy

Figure 2a depicts the XRD diagram of the cA sample, exhibiting the characteristic a-Al_2_O_3_ crystal structure, while a low intensity SiO_2_ peak appears only at the high temperature calcined cA sample at 26.7 2θ, possibly due to crystallization of small amount of SiO_2_ in the Al_2_O_3_ structure. The a-Al_2_O_3_ crystallite size, shown in Table 3, is calculated by applying the Debye-Scherrer formula on the Full Width at Half Maximum (FWHM) of the main peak at 43.4 2θ (Figure 2b).

##### BET Analysis

BET analysis indicated a surface area of 5.32 m^2^/gr for the ncA sample, while no surface area could be measured for the cA, as shown in Table 3. The extended calcination at high temperature (1650 °C) provoked sintering at the nano- and micropores of cA, leading to a dense structure that did not exhibit any nitrogen adsorption and desorption hysteresis during BET analysis, thus the surface area of the sample was not measurable. The pore size distribution diagram for the ncA sample, derived from the nitrogen pressure values during the adsorption step, is shown in Figure 3, and depicts a pore distribution at 50–80 nm. The high temperature calcination modified the porosity of the material by sintering the structure and blocking the original pore network.

##### Scanning Electron Microscopy (SEM)

The microstructural characteristics of the alumina surface may affect the attachment and survival of the cells. Therefore, the morphology of the sample was observed by SEM. The ncA sample at Figure 4a consists of polycrystalline aggregates with a mean size of 400 nm, which sinter with high temperature calcination to larger aggregates in the cA sample (Figure 4b) of 5–20 μm and forms a consistent flat surface.

#### 3.1.2. Wettability

The wettability of the surfaces of non-calcined (ncA) and calcined alumina (cA) substrate was measured by the water contact angle. Values for non-calcined and calcined alumina substrate were 32.6° ± 4.05° and 67.36° ± 4.77°, respectively, which indicates that they have a significantly different wettability (Figure 5), probably due to the elimination of the surface porosity by the calcination procedure.

### 3.2. Biological Assays

#### 3.2.1. Biocompatibility of Al_2_O_3_ Ceramic Substrates with Human NPCs

First, we assessed whether alumina substrates are compatible with human NPCs. Cells were seeded on various matrigel-coated discs, including calcined Al_2_O_3_ and viability was measured. Cells easily attached on most ceramic substrates but a higher viability was observed in Al_2_O_3_ and ZrO_2_ substrates (Appendix A). NPCs were further cultured on Al_2_O_3_ or ZrO_2_ discs in order to measure their relative proliferation. Cells on Al_2_O_3_ almost doubled in number within 6 days, as measured by MTT assay (Figure 6), similar to NPC proliferation on ZrO_2_, (Appendix A) which has been previously utilized for bone tissue engineering in combination with human stem cells [24,25].

SEM indicated that cells formed a layer on the top of the disc covering the whole surface of the substrate (Figure 7a). Then, we asked whether NPCs grown on alumina discs retain their differentiation ability. To do so, cells were spontaneously differentiated towards mixed neuronal populations. As shown in Figure 7b, NPCs formed a neuronal network after three weeks of differentiation. Differentiated cells displayed a high degree of inter-connectivity, suggesting the formation of active synapses.

The efficiency of differentiation was determined by RT-qPCR. Differentiated NPCs expressed high levels of NeuN and TUBB3, two markers of mature neurons and S100B, a glial marker compared to undifferentiated NPCs (Figure 8). This indicates that the neuronal network formed on the top of Al_2_O_3_ discs contains mixed populations of viable neurons and glial cells.

#### 3.2.2. Functionality of the Neuronal Network on Al_2_O_3_ Ceramic Substrates

Next, we asked whether the neuronal network formed on Al_2_O_3_ discs is functional. Functionality of the network was indirectly assessed by expression of pre- and post-synaptic markers in differentiated NPCs, measurement of their long-term viability and observation of cellular morphology. Differentiated cells expressed high levels of synaptophysin (SYP), a pre-synaptic marker, neuroligin-1 (NLGN1), a post-synaptic marker regulating synaptic maturation and glutamate receptor-interacting protein 1 (GRIP1), a post-synaptic marker that binds to glutamate receptors (Figure 9).

Functionality of neurons is directly related to their viability. Therefore, the viability of differentiated NPCs cultured on Al_2_O_3_ discs was estimated. Even though approximately 50% of cells died upon their differentiation and maturation (21 days), a significant number of differentiated NPCs were viable for up to 35 days (Figure 10). This suggests that they remain functional in long-term cultures.

Furthermore, viable cells had short and long neuronal outgrowths (Figure 11) indicating the potential presence of both dendrites and neurites, as those detected in functional neurons. Collectively, these results indicate that the neuronal network formed on Al_2_O_3_ discs contains functional neurons that can efficiently transduce neuronal signals.

## 4. Discussion

Al_2_O_3_ exist in various structures which irreversibly transform into α-alumina upon heat treatment at 1050–1200 °C [26]. α-Alumina is thermodynamically the most stable phase and exhibits high elastic modulus (stiffness), hardness, and excellent resistance to the attack of strong inorganic acids (Table 1) [20], characteristics that have established alumina as a promising material both for tissue engineering implant and stem cell seeding scaffolds. Additionally, porous alumina due to its non-degrading characteristics has a variety of biological applications in vitro and in vivo, including immuno-isolation, biofiltration, or biosensing [27].

As a biomaterial, alumina has been applied in nanoparticle, coated particle, porous membrane and scaffold forms, addressing different biocompatibility issues related to the respective material structural and morphological characteristics [28,29]. cA sample is pristine alumina, exhibiting distinct narrow peaks attributed to the polycrystalline a-Al_2_O_3_ structure, well known for its biocompatibility, which is also in consistency with the attachment and proliferation of the stem cells exhibited in Figure 6 and Figure 7. The BET analysis in Table 3 showed that the cA sample has no porosity, in contrast to the ncA, which had a pore size distribution in the range of 50–70 nm. The morphology by SEM depicted that calcination resulted in the sintering of the ncA aggregates at the range of ~1μm with irregular connectivity and produced larger aggregates of ~5μm with coherent connection, restricting height irregularities at a flatter structure of Figure 4b in contrast to that of Figure 4a, which exhibited cavities in the 1 μm range between the alumina aggregates.

The water contact angle in Figure 5 shows the wettability potential of the samples, denoting the increased wettability of the nCA due to the lower calcination exposure, which is in agreement with the BET analysis. When alumina—as a metal oxide MexOy—is exposed to water or humid air (15% relative humidity), surface interactions with water can form hydroxyl (–OH) groups. These groups facilitate the adsorption of proteins and increase its wettability [30], allowing the attachment and proliferation of cells.

In recent studies, there is evidence for the direct effect of pore size to cell viability, especially in alumina membranes with well-structured hexagonal pore network of 20 nm and 200 nm. This shows that the 20 nm substrate induced lower degree of cell spreading and activation [31] while the 200 nm one showed homogeneous cell adhesion with lack of cell aggregation along the membrane surface. The controlled nano- and microporosity of alumina substrates that can be synthesized by various techniques such as anodic synthesis (PAA) or by slip casting [32,33] can be optimized for bioengineering applications with neuronal cells.

In our study, the homogeneous covering of cA substrates with matrigel in combination with the decreased porosity of the biomaterial promoted the efficient attachment, proliferation and differentiation of human NPCs. Matrigel coating, potentially mediated by the available hydroxyl groups on the surface of cA, mimics the extracellular matrix; it also provides extrinsic signals which are necessary for the survival of neurons and regulate their functionality. Even though the neuronal network formed on cA substrates is functional for a significant period of time, further optimization of the culturing conditions is needed (e.g., by adding extra layers of supporting cells, such as astrocytes) in order to increase the viability of neurons and the long-term functionality of the network. Furthermore, the surface and morphological characteristics of the primary material can be modified to increase its biocompatibility as a scaffold for cell culture, reproducing desired 3D tissue architectures.

Organoids having a 3D spatial distribution of cells are considered developmental models of human organs at a miniature scale [34,35]. They are utilized for disease modeling and offer the possibility to replace animals for functional analysis or drug testing [36]. However, the technology for their generation requires further improvement as they are restricted to a small size and are characterized by significant cell death. Alternatively, organoids can be generated on scaffolds made by biocompatible materials, which can be afterwards seeded with cells. In combination with 3D printing, such scaffolds may be designed to contain channels which would efficiently supply the necessary nutrients to the inner layers of the organoid and increase cell viability (reviewed in [37]). In addition, by employing additive manufacturing methods, favorable properties of 3D sponge or foam porous scaffolds, commonly used in tissue engineering (e.g., bone regrowth, vascularization, and extracellular matrix deposition) can be acquired (e.g., interconnected pore structure for enhanced mechanical properties, uniform tissue development, and rapid mass transport kinetics) [38,39]. To this end, ceramics, including alumina, may provide permanent support compared to biodegradable materials whose degradation might affect cell viability and the 3D architecture of the organoid.

Even though ceramics may not be optimal biomaterials for regenerative medicine in soft tissues (e.g., brain), they have several advantages as they allow the spreading of cells, elongation of their axons, and formation of active synapses. Alumina is a cost-effective biomaterial and can be used for the construction of solid matrixes directing the growth of axons, potentially improving the integration of an implant into neural tissue [5]. Our work also indicates the potential use of alumina for the construction of neural organoids or engineered ex vivo biosystems (e.g., biosensors coated with neural stem cells). Alumina may be a valuable biomaterial for biomedical applications using human neurons. The envisaged application is the development of an organoid mimicking brain function, in which a sponge like structure will ensure the prerequisite properties with its optimized interconnected pore structure. As a next step of ours, additive manufacturing methods will be employed in order to attempt to build an organoid 3D structure based on alumina.

## Figures and Tables

**Figure 1 jfb-11-00065-f001:**
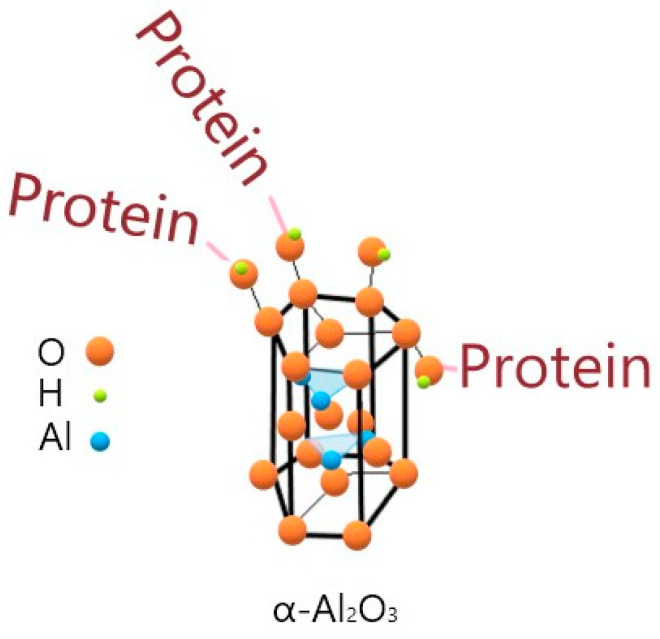
Hexagonal crystal structure of α-Al_2_O_3_ and proposed protein attachment on surface hydroxyl (–OH) bonds.

**Figure 2 jfb-11-00065-f002:**
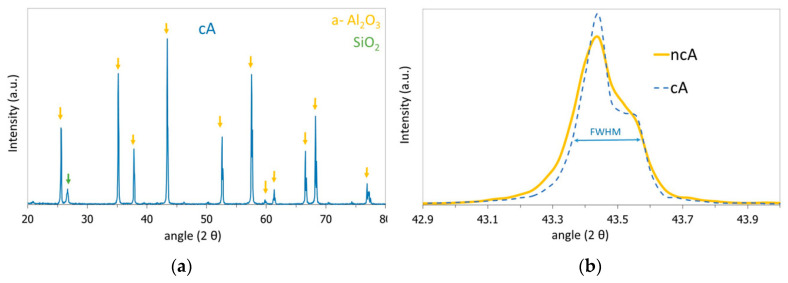
(**a**) XRD diagram of calcined (cA) sample depicting the a-Al_2_O_3_ crystal structure and (**b**) main a-Al_2_O_3_ peak at 43.4 2θ comparison for the two samples.

**Figure 3 jfb-11-00065-f003:**
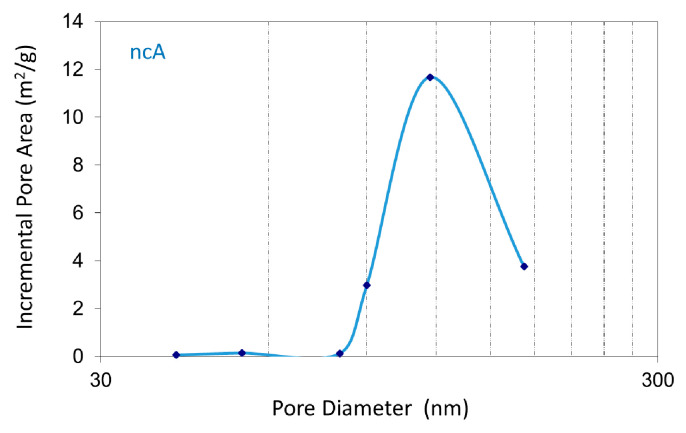
Pore size distribution of the non-calcined (ncA) sample.

**Figure 4 jfb-11-00065-f004:**
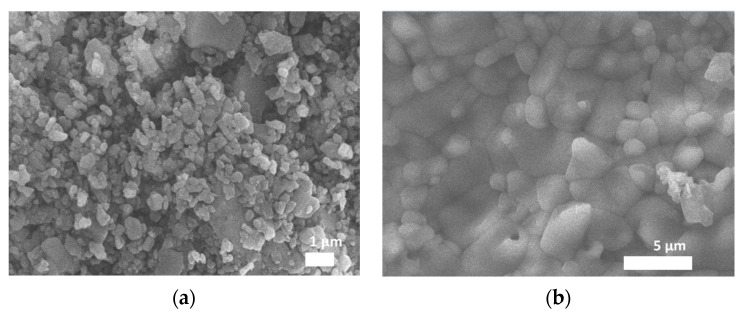
Scanning electron microscopy (SEM) images of (**a**) ncA and (**b**) cA samples.

**Figure 5 jfb-11-00065-f005:**
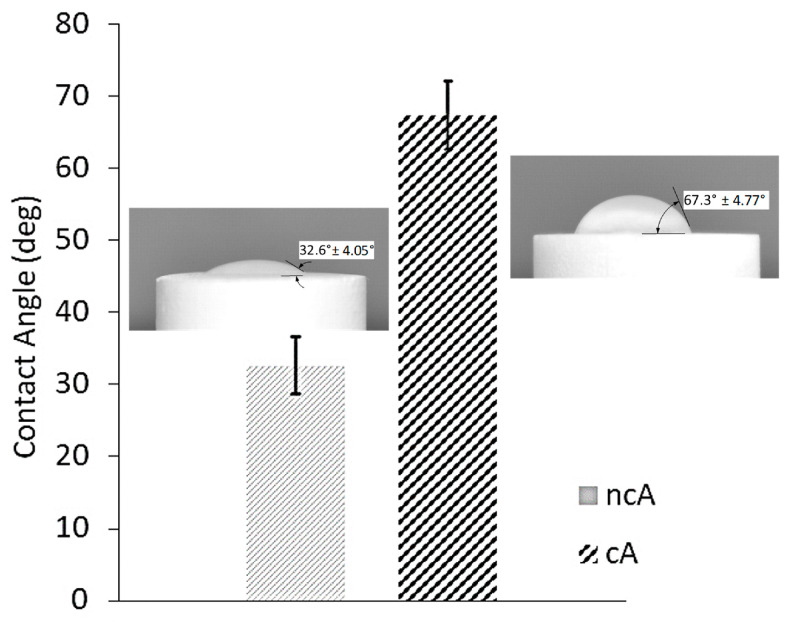
Contact angle values obtained from triplicate drops, highlighting the effect of calcination. Error bars denote ± SD. Inserts are optical images showing contact angle values for 50 µL of dDI water on non-calcined (ncA) or calcined (cA) a-alumina.

**Figure 6 jfb-11-00065-f006:**
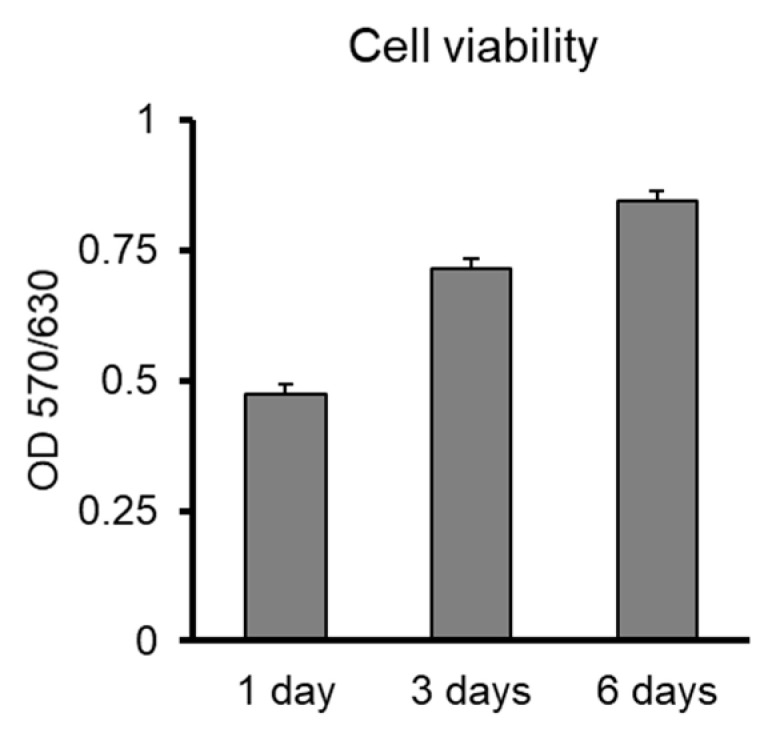
Biocompatibility of Al_2_O_3_ substrates with human NPCs-Viability of cells cultured on calcined matrigel-coated Al_2_O_3_ discs.

**Figure 7 jfb-11-00065-f007:**
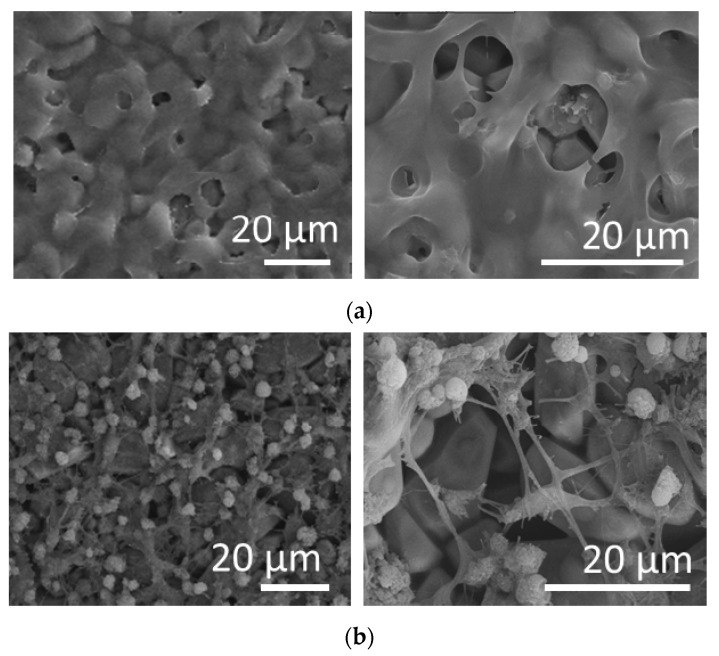
Biocompatibility of Al_2_O_3_ substrates with human NPCs-SEM images of (**a**) undifferentiated and (**b**) differentiated NPCs (scale bar = 20 μm).

**Figure 8 jfb-11-00065-f008:**
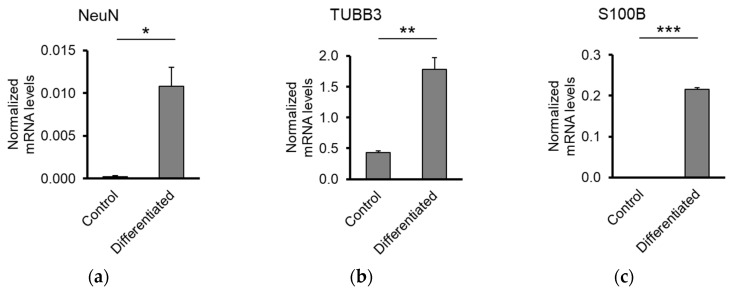
Biocompatibility of Al_2_O_3_ substrates with human NPCs-Expression levels of the neuronal markers (**a**) NeuN; (**b**) TUBB3; (**c**) S100B. Error bars denote ± SD (* *p*-value < 0.05, ** *p*-value < 0.01, *** *p*-value < 0.001).

**Figure 9 jfb-11-00065-f009:**
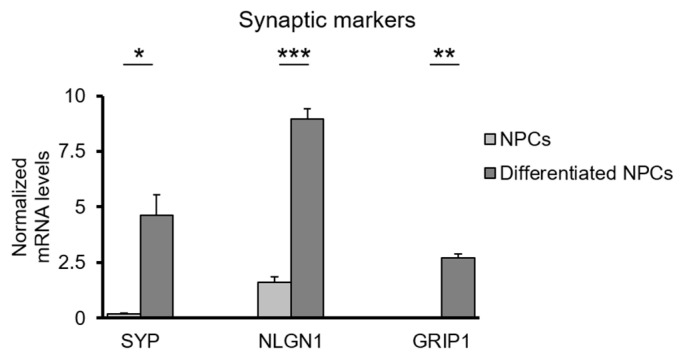
Functionality of NPC-derived neurons on Al_2_O_3_ substrates-Expression levels of synaptic markers in differentiated and undifferentiated NPCs. Error bars denote ± SD (* *p*-value < 0.05, ** *p*-value < 0.01, *** *p*-value < 0.001).

**Figure 10 jfb-11-00065-f010:**
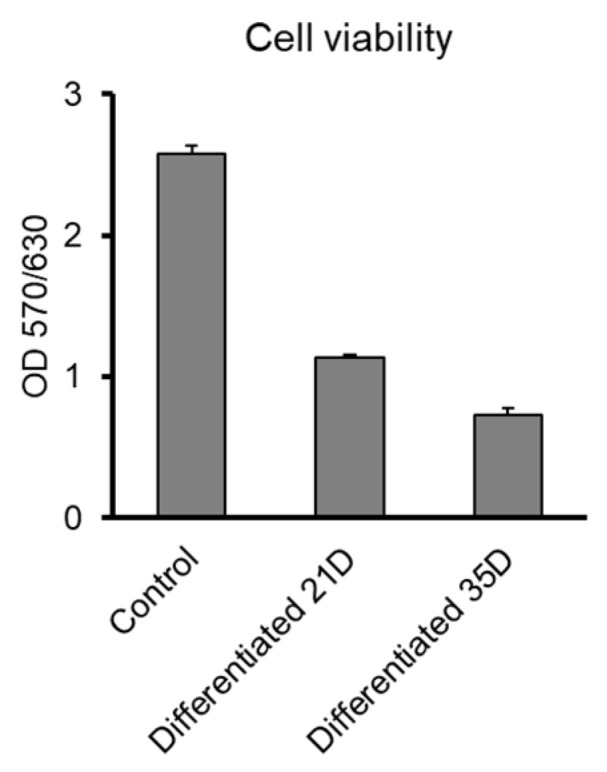
Functionality of NPC-derived neurons on Al_2_O_3_ substrates—Viability of differentiated cells in long-term cultures.

**Figure 11 jfb-11-00065-f011:**
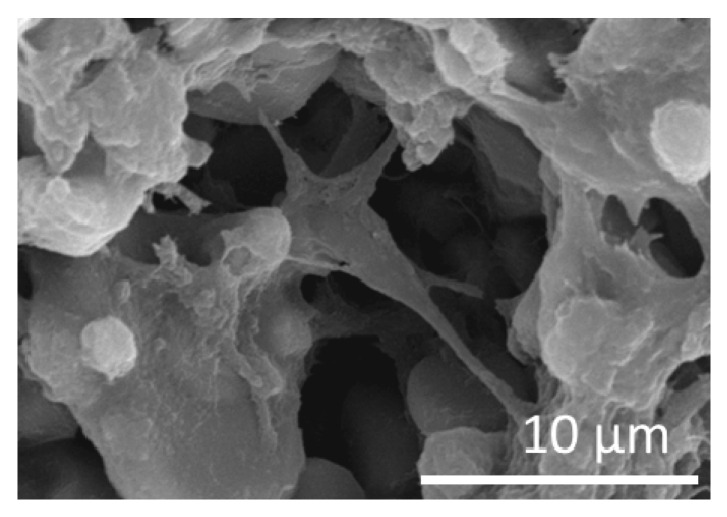
Functionality of NPC-derived neurons on Al_2_O_3_ substrates-SEM image of a viable neuron cultured for 35 days on matrigel-coated Al_2_O_3_ discs (scale bar = 10 μm).

**Table 1 jfb-11-00065-t001:** Properties of Τ-ceramics machinable alumina, THERMANSYS^®^ [20].

Property	Value	Property	Value
Max. Operating Temperature, °C	1750	Dielectric Strength, AC-KV/mm	16.9
Porosity, % vol.	<0.5	Dielectric Constant, 1 MHz	9.8
Density, gr/cm^3^	3.8	Compressive Strength, MPa.	2600
Color	Ivory	Flexural Strength, MPa	380
Thermal Conductivity at 20 °C, W/mK	30	Elastic Modulus, GPa	375
Thermal Conductivity at 800 °C, W/mK	8	Shear Modulus, GPa	152
Coefficient of Thermal Expansion, 10–6/°C	8.4	Hardness, kg/mm^2^	1440

**Table 2 jfb-11-00065-t002:** Sequences of primers used in RT-qPCR experiments.

Target Gene	Forward Primer	Reverse Primer
NeuN	5′-CCAAGCGGCTACACGTCT-3′	5′-GCTCGGTCAGCATCTGAG-3′
TUBB3	5′-CCAAGGGTCACTACACGGAG-3′	5′-ATGATGCGGTCGGGATACTC-3′
S100B	5′-ATGTCTGAGCTGGAGAAGG-3′	5′-CTCATGTTCAAAGAACTCGTG-3′
SYP	5′-CTGTGACCTCGGGACTCAAC-3′	5′-CATAGTCAGGCTGGTAGCCG-3′
NLGN1	5′-CCTTTCCAGCTGGGCTGTTA-3′	5′-TCTGGGGGTCGTCTGGTATT-3′
GRIP1	5′-CCGTTGTCAAATTCTGAGGCG-3′	5′-TACCGTCAGACCCAGGGTAG-3′

**Table 3 jfb-11-00065-t003:** Alumina Sample characteristics.

Sample	Calcination Temperature (°C)	Surface Area (m^2^/gr)	Crystallite Size (nm)
ncA	800	5.56	42.28
cA	1650	-	42.29

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
