# Peer review of "Biocompatibility of α-Al2O3 Ceramic Substrates with Human Neural Precursor Cells"

_jfb, 2020, doi:10.3390/jfb11030065_

Round 1
Reviewer 1 Report
Numéro du manuscrit: jfb 896214 R1
Comment to the authors
Title : Biocompatibility of α-Al2O3 ceramic substrates with human neural precursor cells
The goal of this study is to investigate the biocompatibility of of non‐calcined and calcined alumina disk with human neural precursor cells.
One of the objectives of biomedical research in clinical nanomedicine and nanobiotechnology is to develop new organic / inorganic materials. Among the different forms of materials, three-dimensional (3D) foam materials are very promising candidates to provide conditions mimicking in vivo environments, allowing efficient cell adhesion, proliferation and differentiation due to their unique properties. These include the highest biocompatibility among nanostructures, high surface-to-volume ratio, 3D porous structure (to provide homogeneous / isotropic tissue growth), very favorable mechanical characteristics and rapid mass transport kinetics and electrons (which are needed for the chemistry and physical stimulation of cells).
The article does not describe the choice of alumina for the targeted clinical applications and especially the other materials already commonly used. The authors mention 2 recent publications (ref 12 and 13) but they date from 1991 and 2005.
In recent years alumina has been preclinically and clinically studied in designing dental and orthopedic biomaterials [12,13]
- McLean, J.W. The science and art of dental ceramics. Operative dentistry 1991, 16, 149‐156.
- Swan, E.E.; Popat, K.C.; Desai, T.A. Peptide‐immobilized nanoporous alumina membranes for enhanced osteoblast adhesion. Biomaterials 2005, 26, 1969‐1976.
The authors speak of a "scaffold" structure and porous sample. However, the porosity is very low, the size of the pores very small and above all at no time has the study of the interconnections been approached. In this case, it seems difficult to speak of a 3D porous structure.
Another shortcoming, in the biological study, no control is presented (either another commercial material, thermanox, etc.) and only the increase in the relative proliferation is discussed. Under these conditions, it is essential to take into account a control in order to be able to compare the results.
Finally, the highlights and challenges of non‐calcined and calcined alumina are not compared to the current scaffold competitors. With this control, the results and the discussion would be more interesting.
Author Response
"Please see the attachment."

Reviewer 2 Report
Reviewer comments
- In Page2, Line63. The author described that “In recent years alumina has been preclinically and clinically studied in designing dental and orthopedic biomaterials [12,13]”. However, the above references were published in 1991 and 2005, respectively. Can authors choose more recent articles?
- In Page3, Line115. The sentence was described that “Calcined Al2O3 ceramic discs were coated with Matrigel (BD Biosciences)”. On the other hand, In Page 6, Line186. The author described that “The microstructural characteristics of the alumina surface may affect the attachment and survival of the cells.”. Can author can explain that the good biocompatibility of ceramic substrates is not caused by Matrigel?
- In Page5, Line164. Figure 1 depicts the hexagonal crystal structure of α‐Al2O3 and a proposed mechanism for protein attachment on the surface hydroxyl groups that are formed when the alumina is found in aquatic environments. However, values for calcined alumina substrate was 67.36° ± 4.77° (Page 6, Line195) and show that its surface is hydrophobic. Is there clear evidence to confirm that the surface of the ceramic disc produces hydroxyl groups to facilitate protein adhesion?
Author Response
Dear Editor,
we would like to thank the reviewer for the valuable comments. Please find a below a point-by-point response/additions in the manuscript that improve the quality of our experimental work.
Reviewer 2
- In Page2, Line63. The author described that “In recent years alumina has been preclinically and clinically studied in designing dental and orthopedic biomaterials [12,13]”. However, the above references were published in 1991 and 2005, respectively. Can authors choose more recent articles?
Response: The authors agree with the reviewer’s comment. We have removed reference McLean, J.W. The science and art of dental ceramics. Operative dentistry 1991, 16, 149‐156. and added two recent references
Rahmati, M.; Mozafari, M. Biocompatibility of alumina‐based biomaterials – A review. Journal of Cellular Physiology 2019, 234(4), 3321-3335, doi: 10.1002/jcp.27292
Bain, F.; Novajra, G.; Vitale-Brovarone, C. Bioceramics and scaffolds: a winning combination for tissue engineering. Frontiers in Bioengineering and Biotechnology 2015, 3, 202, doi: 10.3389/fbioe.2015.00202
describing the experimental and clinical use of alumina.
- In Page3, Line115. The sentence was described that “Calcined Al2O3 ceramic discs were coated with Matrigel (BD Biosciences)”. On the other hand, In Page 6, Line186. The author described that “The microstructural characteristics of the alumina surface may affect the attachment and survival of the cells.”. Can author explain that the good biocompatibility of ceramic substrates is not caused by Matrigel?
Response: We have now included Supplementary Figure 2 which shows the attachment and viability of human NPCs in various Matrigel-coated ceramics. As measured by MTT, NPCs preferably attach to and proliferate on alumina, in a similar manner to ZrO2, a ceramic material which has been previously used for human stem cell culture and bone reconstruction. Therefore, these results indicate that the good biocompatibility of alumina is not caused by its coating with Matrigel but is due to its characteristics.
- In Page5, Line164. Figure 1 depicts the hexagonal crystal structure of α‐Al2O3 and a proposed mechanism for protein attachment on the surface hydroxyl groups that are formed when the alumina is found in aquatic environments. However, values for calcined alumina substrate was 67.36° ± 4.77° (Page 6, Line195) and show that its surface is hydrophobic. Is there clear evidence to confirm that the surface of the ceramic disc produces hydroxyl groups to facilitate protein adhesion?
Response: FTIR (Fourier Transform Infra Red) Spectroscopic analysis was performed on calcined (cA) and uncalcined (ncA) samples (added as Supplementary material) in order to evaluate the presence of hydroxyl bonds on material surface. The spectrum of cA, sample on which NPCs attached to, proliferated and were differentiated shows distinct peaks at 3440 cm-1 and 1640 cm-1 exhibiting the presence of –OH and H-OH groups respectively while the 1075 cm-1 peak is attributed to Al-O-H. The current spectrum provides additional evidence for the formation of surface hydroxyl groups.
In the Results Section 3.1.1. Surface topology and nanoporosity the following text was added “FTIR spectroscopy performed on the cA and ncA samples, (Supplementary Figure 1) validated the presence of characteristic surface hydroxyl groups on which proteins may attach.”

Round 2
Reviewer 1 Report
New results have been added. Introduction and discussion are well presented.
The manuscript is accepted in present form.